# A Robust Semi-Direct 3D SLAM for Mobile Robot Based on Dense Optical Flow in Dynamic Scenes

**DOI:** 10.3390/biomimetics8040371

**Published:** 2023-08-16

**Authors:** Bo Hu, Jingwen Luo

**Affiliations:** School of Information Science and Technology, Yunnan Normal University, No. 768 Juxian Street, Chenggong District, Kunming 650500, China; hb431368@163.com

**Keywords:** dynamic scenes, mobile robot, simultaneous localization and mapping (SLAM), semi-direct method, dense optical flow, relocation

## Abstract

Dynamic objects bring about a large number of error accumulations in pose estimation of mobile robots in dynamic scenes, and result in the failure to build a map that is consistent with the surrounding environment. Along these lines, this paper presents a robust semi-direct 3D simultaneous localization and mapping (SLAM) algorithm for mobile robots based on dense optical flow. First, a preliminary estimation of the robot’s pose is conducted using the sparse direct method and the homography matrix is utilized to compensate for the current frame image to reduce the image deformation caused by rotation during the robot’s motion. Then, by calculating the dense optical flow field of two adjacent frames and segmenting the dynamic region in the scene based on the dynamic threshold, the local map points projected within the dynamic regions are eliminated. On this basis, the robot’s pose is optimized by minimizing the reprojection error. Moreover, a high-performance keyframe selection strategy is developed, and keyframes are inserted when the robot’s pose is successfully tracked. Meanwhile, feature points are extracted and matched to the keyframes for subsequent optimization and mapping. Considering that the direct method is subject to tracking failure in practical application scenarios, the feature points and map points of keyframes are employed in robot relocation. Finally, all keyframes and map points are used as optimization variables for global bundle adjustment (BA) optimization, so as to construct a globally consistent 3D dense octree map. A series of simulations and experiments demonstrate the superior performance of the proposed algorithm.

## 1. Introduction

Simultaneous localization and mapping (SLAM) has gradually become a frontier research hotspot in the field of intelligent robots. SLAM enables the robot to use the environmental information perceived by its sensors to estimate its pose in real time and incrementally construct an environment map. In recent years, with the widespread application of depth cameras, it has been possible to simultaneously obtain both the color and depth information of the environment, i.e., RGB-D information, which provides richer environmental perception for the SLAM system of mobile robots. Thus, visual SLAM (VSLAM) using RGB-D information has rapidly developed [1,2,3,4].

The solutions of VSLAM mainly include feature-based and direct methods. Specifically, the feature-based method relies heavily on the feature points extracted from the image and is sensitive to image quality. In order to improve the robustness of the algorithm in cases of missing features or blurred images, the direct method has emerged to solve the camera’s motion by directly comparing the pixel colors of the images; it avoids feature point extraction and fully utilizes all the information in the image. A typical case is the large-scale direct monocular (LSD) SLAM based on grayscale images, known as LSD-SLAM [5], which can achieve semi-dense reconstruction of complex scenes using the CPU. However, LSD-SLAM is very sensitive to the camera’s intrinsic parameters and exposure, and it is easy to lose tracking when the camera moves quickly. Moreover, due to the inconvenience of implementing loop closure detection based on direct methods, LSD-SLAM needs to rely on feature points for loop closure detection. After this, to enhance the efficiency of the algorithm, a semi-direct monocular visual odometry (SVO) SLAM scheme [6] was proposed, which only tracks sparse feature points. However, due to the inadequate performance of deep filtering in SVO, it possesses poor robustness and requires repeated relocalization.

It is noticed that most of the existing SLAM algorithms are designed mainly for static scenes. Unfortunately, in practical applications, dynamic environmental conditions present significant challenges to robot perception and decision-making behavior. If the feature points of dynamic objects are included in robot pose estimation, it will result in erroneous feature matching, leading to severe trajectory drift and the inability to construct consistent and accurate environmental maps. Although outlier removal algorithms such as random sample consensus (RANSAC) [7] can eliminate some dynamic feature points as outliers, these algorithms still fail when dynamic objects occupy most of the robot’s field of view. Therefore, reducing the impact of dynamic objects in the scene on robot pose estimation is crucial for high-performance and robust 3D SLAM.

In summary, to meet the application requirements of mobile robots in dynamic scenes, this paper proposes a robust semi-direct 3D SLAM algorithm based on dense optical flow. Compared with the feature-based method, we adopt the semi-direct method in the front-end to quickly obtain the initial pose of the camera and use dense optical flow to detect the dynamic areas in the images, which ensures the real-time performance of the algorithm. Meanwhile, dense optical flow is employed to detect the dynamic regions of the image and eliminate map points in the dynamic regions. It should be noted that the dynamic objects in the scene can be accurately detected using the dense optical flow, which in turn obtains the static map points with high confidence and is utilized in the back-end of the system for the optimization of the camera pose, effectively suppressing the interference of the moving objects and improving the robustness of the system while ensuring the accuracy. Then, high-quality keyframes are selected and ORB feature points are extracted from them for mapping and optimization in the back-end. Considering the tracking failures that occur in dynamic scenes, ORB feature points are extracted from the current frame to recover the robot pose by matching them with keyframes and map points. Finally, global optimization is applied to the mobile robot pose and sparse map points, and a 3D dense octree map that can be used for robot navigation and obstacle avoidance is constructed.

The main contributions of this work are as follows:

(1) A high-precision dynamic object detection method based on dense optical flow is proposed, which can accurately determine the dynamic region of the image.

(2) A high-performance keyframe selection strategy is proposed, reducing the influence of dynamic objects on the quality of keyframes and improving the accuracy of the algorithm.

(3) Aiming at the problem that it is difficult to relocate after the direct method tracking loss, an effective relocation method is developed by introducing feature point extraction and matching, hence improving the success rate of relocation and the robustness of the algorithm.

The overall structure of this paper is as follows: Section 2 details the system framework, improvement strategies, and implementation process of our algorithm. Section 3 provides typical experimental results and comparative analyses, which include a series of simulations with a public dataset and a case study with a mobile robot in a real scene. Section 4 summarizes the work and proposes future research directions.

## 2. Related Work

To reduce the impact of dynamic objects in the scene on the robot’s pose estimation, it is necessary to detect the dynamic objects in the visual odometry at the front-end and remove their corresponding 3D points generated by these objects. So far, the dynamic detection schemes used in the front-end of SLAM can be roughly divided into three categories: optical flow-based, geometric-based, and deep learning-based methods.

### 2.1. Optical Flow-Based Methods

As a method for estimating the motion of image pixels, the optical flow-based method has been widely used in dynamic object segmentation. Derome et al. [8,9] used the residuals between different images to calculate the optical flow and detect dynamic objects through the residuals; Wang et al. [10] segmented dynamic objects in the scene by calculating the dense optical flow of the image and clustering the image using sparse point trajectories. This method has good detection performance for rigid and non-rigid moving objects in the scene, but it cannot work in real time; Sun et al. [11] built a dense point cloud map by combining dense optical flow with the CodeBook model to segment the foreground and background, thereby reducing the impact of dynamic objects; Zhang et al. [12] applied the optical flow residuals to dynamic segmentation, namely, first using the PWC-Net [13] to calculate the optical flow of RGB-D images, then combining the optical flow with camera motion to calculate the 2D scene flow, using the scene flow for dynamic segmentation, and, finally, completing the reconstruction of the static background through multiple iterations. The use of the dense optical flow provides better results for dynamic region detection, but dense optical flow needs to calculate the motion of all pixel points in the image, which has a greater impact on the real-time performance of the SLAM algorithm. Likewise, the calculation of optical flow is also highly susceptible to the influence of motion.

### 2.2. Geometric-Based Methods

The geometry-based method distinguishes dynamic features from static features mainly by means of epipolar geometry constraints. Li et al. [14] proposed a static point weighting method for keyframe edge points, which used weight to indicate the possibility that a point belongs to a static point, thereby reducing the impact of feature points generated by dynamic objects on pose estimation; Yuan et al. [15] determined dynamic objects in a scene by combining point features with line features; Dai et al. [16] introduced Delaunay triangulation to establish the connection between different points and separated dynamic feature points from static feature points by the correlation between points; Wei Tong et al. [17] combined polar geometric constraints and the superpixel segmentation algorithm to judge dynamic objects, but the segmentation algorithm reduced the real-time performance of the system; Ai Qinglin et al. [18] classified and filtered the feature points by strict geometric constraints and finally estimated the pose and constructed the map by static feature points only. The geometry-based based method has high real-time performance, but it struggles to deal with non-rigid dynamic scenes.

### 2.3. Deep Learning-Based Methods

Most recently, deep learning-based dynamic detection methods have shown high accuracy. The DS-SLAM proposed by Yu et al. [19] employed a semantic segmentation network to filter out dynamic objects in the scene and then calculated the precise pose of the camera based on the remaining static points; Yan et al. [20] performed dynamic segmentation by combining motion residual information of adjacent frames with the YOLACT++ [21,22] instance segmentation network, and the results showed high segmentation accuracy; DynaSLAM [23] segmented a priori dynamic objects in the scene by adopting the instance segmentation network Mask R-CNN [24] and utilized multi-view geometry to detect potential dynamic objects, which improved the accuracy of the SLAM system to detect dynamic objects. Unfortunately, these deep learning-based approaches need a priori information and have the characteristic of high computational complexity, making them difficult to run in real time.

## 3. Algorithm Framework

The overall framework of our scheme is shown in Figure 1. For the front-end, the sparse direct method is utilized to preliminarily estimate the pose of the robot, and the first frame of the RGB-D image acquired by the depth camera is used as the initial keyframe. The initial pose of the robot is calculated by minimizing the photometric error, and the correspondence of pixels is optimized by feature alignment to obtain the matching points of adjacent frames. Then, the dense optical flow field of two adjacent frames is calculated to judge the dynamic area. In order to reduce the computational complexity of the dense optical flow, down-sampling processing is performed on the image. Meanwhile, to reduce the influence of robot motion on optical flow calculation, the dense optical flow of two adjacent frames is calculated by combining the homography matrix. By setting a dynamic threshold to separate dynamic regions in the scene, the points in the dynamic area in the current frame are eliminated, and the remaining static points are chosen to further optimize the robot’s pose. After successful tracking of the robot’s pose, a new keyframe is selected and the ORB feature points are extracted in its static area. 

For the back-end, the keyframe’s pose and map points continue to be optimized by local bundle adjustment (BA). If the robot poses tracking fails, relocation is performed using the feature point method, where ORB feature points are extracted in the current frame and matched with the previous keyframes or map points in the map to recover the pose. In this way, all keyframes and map points are then employed as optimization variables for global BA optimization, resulting in a globally consistent robot trajectory and 3D dense map. Finally, the 3D navigation map is generated by coupling with the octree map.

### 3.1. Mobile Robot’s Pose Tracking

In our work, after obtaining an RGB-D image with the depth camera, the sparse direct method is used to preliminarily estimate the robot’s pose, and the initial pose of the robot is computed by minimizing the photometric error between the corresponding pixel points of the same 3D point in adjacent frames. When the previous frame is successfully tracked, the constant velocity motion model is exploited to predict the pose of the current frame, and the 3D points tracked in the previous frame are projected to the current frame by the estimated pose. Based on the assumption of photometric constancy, the photometric error of the pixel points corresponding to the same 3D point between the adjacent frames *I_k_*, *I_k_*_−1_ is as follows:(1)$$\delta I\left(T,u\right)=I_{k}\left(\pi \left(T\cdot \pi^{-1}\left(u\right)\right)\right)-I_{k-1}\left(u\right)$$
where *T* is the robot pose, *u* is the position of the pixel point in the previous frame, and *π* is the projection function. 

However, in practical applications, there may be a photometric difference between two adjacent frames caused by factors such as illumination, shadow, exposure, etc. Therefore, to reduce the photometric difference between adjacent frames, this paper introduces two photometric compensation variables *α* and *β* to perform inter-frame photometric compensation on the previous frame image, i.e.,
(2)$$\delta I\left(T,u\right)=I_{k}\left(\pi \left(T\cdot \pi^{-1}\left(u\right)\right)\right)-\left(I_{k-1}\left(u+\alpha \right)+\beta \right)$$
where *α* is the photometric gain coefficient, and *β* is the offset of photometric.

By minimizing the photometric error between adjacent frames, the rough pose estimate *T_k_*_,*k*−1_ from the previous frame to the current frame can be obtained as follows:(3)$$T_{k,k-1}=\underset{T_{k,k-1}}{\arg\min}\frac{1}{2}\sum_{i\in R}^{}{}_{}{\left\|\delta I\left(T_{k,k-1},u_{i}\right)\right\|}^{2}$$
where *u_i_* is the *i*th pixel in the previous frame, *R* is the whole image.

For a general case, only using adjacent frames to calculate the robot’s pose is prone to cumulative drift, so it is necessary to search for local maps to obtain more map points that need to be tracked and optimized. Specifically, the local map is first tracked to obtain the covisibility keyframes and covisibility map points of the current frame, and then the covisibility map points are projected to the current frame using the poses obtained in the previous step. For each successfully projected map point, the correspondence between the pixel points and map points of the current frame is optimized by minimizing the photometric error of the corresponding pixel points.

As shown in Figure 2, in order to ensure the uniform distribution of the projected points, the current frame image is divided into a grid of 5 × 5, and the covisibility keyframe with the shortest distance to the current frame is set as the reference keyframe of the current frame. Then, by tracking the covisibility map points between the reference keyframe and the current frame, the corresponding map points in the reference frame are projected onto the current frame. Further, the grid is divided according to the current frame, and then a maximum of 5 map points are selected within each grid for pixel point matching. Owing to the inaccuracy of the preliminary pose estimation, the projected position of the map point in the current frame has a certain error compared to the real position. According to the photometric consistent assumption, the photometric value of the pixel points corresponding to the same map point in the reference keyframe and the current frame is consistent. Thus, the correspondence of pixel points can be optimized by minimizing the photometric error as follows:(4)$$u_{i}{}^{\prime }=\arg\underset{u_{i}}{\min}\frac{1}{2}{\left\|I_{k}\left({u^{\prime }}_{i}\right)-I_{r}\left(u_{i}\right)\right\|}^{2},\forall i$$
where *u_i_′* and *u_i_* denote the position of the pixel point in the current frame and the covisibility keyframe, respectively.

Since there may be scale problems caused by the distance and angle between the covisibility keyframe and the current frame, the two frames cannot be aligned directly, so the affine matrix *A_i_* [6] is introduced and the pixel points in the reference keyframe are affine warped and then aligned with the current frame: (5)$$u_{i}{}^{\prime }=\arg\underset{u_{i}}{\min}\frac{1}{2}{\left\|I_{k}\left(u_{i}{}^{\prime }\right)-A_{i}I_{r}\left(u_{i}\right)\right\|}^{2},\forall i$$

Further, the Gauss–Newton method is adopted to solve the above equation to optimize the correspondence of pixel points.

### 3.2. Dynamic Region Detection

The main idea of optical flow is based on the photometric consistency assumption between adjacent frames [25], and the optical flow equation can be written as follows:(6)$$I_{}\left(x,y,t\right)=I_{}\left(x+dx,y+dy,t+dt\right)$$
where *I*(*x*, *y*, *t*) denotes the photometric function of the pixel point (*x*, *y*), and (d*x*, d*y*) denotes the distance that the pixel point (*x*, *y*) moves within time step d*t*. Assuming that *f_x_* and *f_y_* are the horizontal optical flow amplitude and vertical optical flow amplitude of the pixel point, respectively, we obtain the following:(7)$$f_{x}=\frac{dx}{dt},f_{y}=\frac{dy}{dt}$$

To reduce the impact of dynamic objects on the robot’s pose estimation, this paper combines dense optical flow for dynamic detection and eliminates dynamic points in dynamic regions. The remaining static points are used to further optimize the robot’s pose. The dense optical flow method determines the dynamic regions in the image by calculating the motion of each pixel in the image. Compared with the sparse optical flow, the dense optical flow can provide more detailed motion information, but the dense optical flow is more computationally intensive. Scenarios for the dynamic areas’ detection is shown in Figure 3.

The traditional optical flow is based on the assumption of a static background and can effectively track dynamic objects when the mobile robot is stationary or has a small displacement. However, when the mobile robot moves in translation or rotation, the image will undergo deformation, resulting in the static object and the background generating optical flow as well, which may interfere with the detection of dynamic objects. To reduce the impact of the motion of the mobile robot on the optical flow calculation, this paper adopts the homography matrix to compensate the image. The homography matrix characterizes the correspondence of 3D points on the same plane at different viewpoints, which can correct the image deformation caused by the motion of the mobile robot. Equation (8) represents the coordinate correspondence before and after the homography transformation, and *H_t+1_* is the homography matrix from frame *t* to frame *t* + 1.
(8)$$\left[\begin{array}{c} x_{t+1}\\ y_{t+1}\\ 1 \end{array}\right]=H_{t+1,t}\;\left[\begin{array}{c} x_{t}\\ y_{t}\\ 1 \end{array}\right]$$

In this paper, the correspondence between pixel points is calculated by minimizing the photometric error to optimize the correspondence of pixel points, so that the matching points between adjacent frames can be obtained, and then the homography matrix of two adjacent frames can be obtained by four pairs of matching points. However, on account of the excessive number of matching points in adjacent frames and a large number of mismatches, the RANSAC algorithm is employed to solve the homography matrix to obtain more robust results. Then, the homography matrix is further used to compensate for the current frame image, and the optical flow field is computed with the previous frame image. This approach effectively mitigates optical flow interference caused by camera motion and enhances the accuracy of solving the optical flow field. 

Typically, dense optical flow requires calculating the motion of each pixel in the image, which consumes a large amount of computing resources. In order to improve the efficiency and real-time performance of the algorithm, this paper adopts down-sampling the image as a means of improving algorithm efficiency and real-time performance. After obtaining the image *I_t_* at time *t* and the image *I′_t_*_+1_ after homography compensation at time *t* + 1, down-sampling is performed on the two frames to reduce their resolution. In order to ensure that the details and clarity of the original image are retained after the down-sampling, the bilinear interpolation [26] method is utilized for down-sampling, as shown in the following:(9)$$J\left(i,j\right)=\frac{\sum_{k=1}^{4}\left(x_{k}-x\right)\left(y_{k}-y\right)p_{k}}{\left(x_{2}-x_{1}\right)\left(y_{2}-y_{1}\right)}$$
where *J*(*i*, *j*) denotes the value of the (*i*, *j*)th pixel point of the down-sampled image, *p_k_* denotes the value of the *k*th pixel point in the original image, (*x*, *y*) denotes the corresponding coordinates of the pixel points in the original image, and (*x_k_*, *y_k_*) denotes the coordinates of the 4 nearest pixels to (*x*, *y*).

Then, the dense optical flow of images *I_t_* and *I′_t_*_+1_ before and after down-sampling is calculated to obtain a low-resolution optical flow field. Finally, the low-resolution optical flow field is up-sampled again to recover to the resolution of the original optical flow field image. Although this processing may reduce the accuracy of optical flow calculations to a certain extent, it can significantly improve the time of dense optical flow calculation.

After obtaining the optical flow field of the original resolution image, in order to distinguish between dynamic and static regions in the image, the optical flow amplitude *f* is calculated for each pixel point (*x*, *y*) based on its horizontal optical flow amplitude *f_x_* and vertical optical flow amplitude *f_y_*, i.e.,
(10)$$f=\sqrt{f_{x}{}^{2}+f_{y}{}^{2}}$$

In general, the magnitude of optical flow amplitude *f* can be used to represent the displacement of a pixel point between two frames. The optical flow amplitude *f* of each pixel point (*x*, *y*) is compared with a set dynamic threshold *f*th, and if the optical flow amplitude *f* > *f*th, the pixel point is considered to have a large displacement between two frames and is marked as a dynamic pixel point *P_dynamic_*; otherwise, it is a static pixel point *P_static_*, as shown in Equation (11). In the subsequent optimization process, the points falling in the dynamic region are eliminated to improve the efficiency and accuracy of the optimization.
(11)$$\left(x,y\right)\in \left\{\begin{array}{l} P_{dynamic},\;\sqrt{f_{x}{}^{2}+f_{y}{}^{2}}>f_{th}\\ P_{static},\;otherwise \end{array}\right.$$

As a note, the setting of dynamic threshold *f*th is based on the rotation and translation of the robot, so its expression can be written as follows:(12)$$f\mathrm{th}=\alpha \frac{{\left\|T_{rw}T_{cw}{}^{T}\right\|}_{F}}{{\left\|t\right\|}_{2}}$$
where *T_rw_∈SE(3)* is the transformation matrix of the reference keyframe from the world coordinate system to the camera coordinate system, *T_cw_∈SE(3)* is the transformation matrix of the current frame from the world coordinate system to the camera coordinate system, *T_rw_T_cw_^T^* describes the rotation of the camera, ‖·‖*_F_* is the Forbenius norm, defined as the square root of the sum of the squares of the absolute values of all elements of the matrix, *t* is the translation of the camera vector, ‖·‖_2_ is the Euclidean norm, defined as the squared sum of the absolute values of all elements of the vector and recalculated, and *α* is the scale factor. 

As the calculation of optical flow is affected by camera rotation and translation, when the range of the robot’s motion is large, the dynamic threshold *f*th increases with the increase in the transformation matrix of the current frame and the reference frame, so as to filter out the optical flow noise caused by rotation; when the robot is panning or stationary, the dynamic threshold *f*th decreases with the increase in the translation vector, allowing the robot to detect some small dynamic objects in the scene during motion.

Figure 4 compares the results of different optical flow detection methods, where Figure 4a shows the original input image with a pedestrian in motion in the scene, while the camera is also moving to the right. From Figure 4b, it can be seen that the sparse optical flow only detects a few dynamic feature points on the pedestrian, and some of the feature points in the background are also mistakenly detected as dynamic points. In Figure 4c, the red area represents the detected dynamic areas and we observed that although pedestrians are detected as dynamic objects, the background and stationary objects are also mistaken for dynamic regions because of the camera motion. We could reasonably conclude from Figure 4d that our method removes most of the false detections and only detects the pedestrian part as a dynamic area, indicating that the use of homography matrix compensation effectively reduces the impact of camera motion on optical flow and the dynamic threshold still provides good detection results for optical flow during camera rotation and rapid motion, thus improving the robustness of optical flow calculation.

### 3.3. Mobile Robot’s Pose Optimization

Through the above processing, the dynamic region information of the current frame can be well determined. To reduce the impact of dynamic objects on robot pose estimation, the points falling in the dynamic area of the current frame are marked as dynamic points and eliminated, while the map points that successfully match with the points in the dynamic area of the current frame are also considered as outliers and removed from the local map. After removing the dynamic points, the feature points set *u_s_* that matches the static map points is obtained and, by minimizing the reprojection error, the robot’s pose is optimized again to reduce the impact of dynamic objects on the robot’s pose, i.e.,
(13)$$T_{k,w}=\arg\underset{T_{k,w}}{\min}\frac{1}{2}\sum_{u_{i}\in u_{s}}{\left\|u_{i}-\pi \left(T_{k,w}P_{i}\right)\right\|}^{2}$$

### 3.4. Keyframe Selection

In our case, the front-end of the algorithm performs pose tracking through direct methods, and the subsequent steps such as loop closure detection, relocation, and mapping are all based on keyframes. Therefore, the reasonable selection of keyframes can enhance the efficiency of the algorithm and lower the memory usage of the system. At present, most SLAM algorithms select keyframes by mainly focusing on the time interval from the last keyframe insertion, the number of feature points in the current frame, and the number of keyframes in the system, without considering the influence of camera motion and dynamic objects in the scene. Therefore, drawing on the conventional keyframe selection strategy, we add the pose transformation of the robot and the optical flow amplitude of the current frame as the condition of keyframe selection.

First, the rotation matrix *R_rc_* of the current frame relative to the reference frame is calculated by using the rotation matrix of the current frame and the reference keyframe relative to the world coordinate system, i.e.,
(14)$$R_{cr}=R_{rw}R_{cw}^{T}$$
where *R_cw_* and *R_rw_* denote the rotation matrix of the current frame and the reference keyframe relative to the world coordinate system, respectively. Furthermore, the rotation angle Δ*α_i_* of the robot can be computed by the rotation matrix *R_cr_* as follows:(15)$$\Delta \alpha_{i}=\arccos\left(\frac{tr\left(R_{cr}\right)-1}{2}\right)$$
where *tr*(·) is the trace of the matrix and defined as the sum of the diagonal elements of the matrix. 

Then, the translation Δ*t_cr_* of the current frame relative to the reference frame is established by the translation vector of the current frame *t_cw_* and the translation vector of the reference frame *t_rw_*, that is:(16)$$\Delta t_{cr}=t_{cw}-t_{rw}$$

According to the rotation angle Δ*α_i_* and the translation Δ*t_rc_*, the robot’s pose transformation *V* could be expressed as follows:(17)$$V=\left(1-\beta \right){\left\|\Delta t_{cr}\right\|}_{2}+\beta \left(w\cdot \frac{\Delta \alpha_{i}}{\pi }\right)$$
where ‖·‖_2_ is the Euclidean norm, *w* is the scale factor, and *β* is the weight of camera rotation angle. For a set threshold *V*th, *V* > *V*th means that the current camera motion is larger, which may lead to difficulty in feature point matching or tracking loss. Thereby, the current frame needs to be selected as a keyframe to avoid tracking failure or pose drift.

Next, the optical flow amplitude is exploited to determine whether to select a keyframe. More specifically, in the dynamic region detection stage, the optical flow amplitude *f_i_* of each pixel in the image is obtained, and then the variance *σ*^2^ of the optical flow amplitude of all pixels in the whole image is calculated to indicate the degree of image variation in that frame. Thus, we obtained the following: (18)$$\sigma^{2}=\frac{1}{N}{\sum_{i=1}^{N}\left(f_{i}-\frac{1}{N}\sum_{i=1}^{N}f_{i}\right)}^{2}$$
where *N* is the total number of pixels in the image. If the variance of the optical flow amplitude of a certain frame is relatively large, it means that the motion of some pixel points in that frame deviates from the average motion and there may be the influence of dynamic objects.

Additionally, the ratio *σ* between the optical flow amplitude variance *σ_c_*^2^ of the current frame and the optical flow amplitude variance *σ_r_*^2^ of the previous keyframe is calculated, i.e.,
(19)$$\sigma =\frac{\sigma_{c}^{2}}{\sigma_{r}^{2}}$$

For a set threshold σth, if *σ* > σth, the current frame has undergone significant changes compared to the previous keyframe, which may be influenced by dynamic objects, and thus the current frame is not suitable as a keyframe. In this case, we should skip the current frame and continue to judge subsequent frames. This strategy can effectively reduce the influence of dynamic objects on keyframes, enhance the stability and reliability of keyframes, and decrease the redundancy of keyframes. After inserting a new keyframe, ORB feature [27] extraction is performed in the static area of the keyframe, and then new map points are constructed based on the feature matching relationship between the keyframe and its covisibility keyframe. Next, the keyframes and map points are filtered to remove some low-quality or redundant keyframes and map points. Finally, local BA is utilized to optimize the keyframe pose and map points, resulting in a more accurate and stable local map.

### 3.5. Relocation and Global Optimization

Considering uncertain situations, e.g., motion blur and illumination changes in dynamic scenes, the direct method may fail to track the pose of mobile robots. In this case, the relocation algorithm is needed to recover the robot’s pose. For the direct method, when the tracking fails, the photometric information of the image cannot be used for relocation. Thus, this paper introduces the feature point method for relocation and determines whether to directly match the map points to restore the pose based on the number of selected candidate keyframes. The relocation process is illustrated in Figure 5, and its procedure is listed in Algorithm 1.
**Algorithm 1:** Relocation algorithm**input:** current frame *I_cur_*, keyframe *Kf*, local map *Map*
**output:** camera pose *pose*
1 **if** Tracking → Lost **then**
2  *p_i_* ← Extraction(*I_cur_*)
3  *BowV* ← Compute (*p_i_*)
4  *I_cand_* {*Kf_i_*|*i* = 1…*N_cand_*} ← Detect (*Kf*, *BowV*)
5  **if** *n* > 2 **then**
6   **for** *i* = 0 →*N_cand_* **do**
7    *N_i_* ← Matching(*I_cur_*, *Kf_i_*)
8    **if** *N_i_* > 15
9   **end for**
10   *pose* ← PnPsolver(*N_i_*)
11  **Else**
12   update(*Map*)
13   *M* = {*m_i_*| i = 1…*n* } ← Search(*BowV*, *Map*)
14  *N_i_* ← Matching(*I_cur_*, *m_i_*)
15  **if** *N_i_* > 15
16    *pose* ← PnPsolver(*N_i_*)
17   **if** *N_in_* > 50
18   relocalization ← Optimize(*pose*)
19 end if
20 end if

After the front-end experiences tracking loss, we first extract ORB features from the current frame *I_cur_* that has failed to track and calculate its bag-of-words [28] vector *v_c_* = {*v_ci_*|*i* = 1…*n*}. Then, the L1 distance of the bag-of-words vector is chosen to calculate the similarity score *s* between the current frame and its covisibility keyframes, i.e.,
(20)$$s\left(v_{c},v_{k}\right)=\frac{1}{2}\sum_{i}\left\{\left|v_{ci}\right|+\left|v_{ki}\right|-\left|v_{ci}-v_{ki}\right|\right\}$$
where *v_k_* is the bag-of-words vector of covisibility keyframes, *v_ci,_* and *v_ki_* are the weights of the *i*th word in the current frame and keyframe, respectively. 

In terms of the ranking of keyframe similarity scores from highest to lowest, several keyframes *Kf* with the highest scores are selected as candidate keyframe sets *I_cand_* _=_ {*Kf_i_*|*i* = 1…*n*}. Then, all the candidate keyframes are traversed, the feature points between the current frame *I_cur_* and all the candidate keyframes *Kf_i_* are quickly matched using the bag-of-words vector, and the effective number of matches *N_i_* is filtered according to the quality of the matching points. If the effective number of matches of a candidate keyframe is greater than 15, it is set as the reference keyframe, and the EPnP [29] algorithm is employed to solve the relative poses of the reference keyframe and the current frame. Namely, the 3D points are represented as a combination of 4 control points, and only the 4 control points are optimized. The coordinates of 3D points in the world coordinate system and camera coordinate system are represented in the form of control point coordinates, which are respectively modeled as follows:(21)$$\left\{\begin{array}{l} p_{i}^{w}=\sum_{j=1}^{4}a_{ij}c_{j}^{w}\\ p_{i}^{c}=\sum_{j=1}^{4}a_{ij}c_{j}^{c} \end{array}\right.$$
where $p_{i}^{w}$ and $p_{i}^{c}$ are the coordinates of the *i*th 3D point in the world coordinate system and the camera coordinate system, respectively, $c_{j}^{w}$ and $c_{j}^{c}$ are the coordinates of the *j*th control point in the world coordinate system and the camera coordinate system, respectively, and *α_ij_* is a barycentric coordinate, which can be obtained according to the projection principle of the camera as follows:(22)$$\left[\begin{array}{l} u_{i}\\ 1 \end{array}\right]=Kp_{i}^{c}=\frac{1}{w_{i}}K\sum_{j=1}^{4}a_{ij}c_{j}^{c}$$
(23)$$c_{j}^{c}=Rc_{j}^{w}+t$$
where *w_i_* is the depth value and *u_i_* is the 2D coordinate corresponding to $p_{i}^{c}$. 

By solving Equation (23), the coordinates $c_{j}^{c}$ of the control point in the camera coordinate system can be obtained, thereby obtaining the coordinates $p_{i}^{c}$ of the 3D point in the camera coordinate system. Then, the corresponding relationship between the 3D points in the world coordinate system and the camera coordinate system of the current frame is used to solve the pose of the current frame relative to the keyframe, and the Levenberg–Marquardt method [30] is utilized for optimization, i.e.,
(24)$$T=\arg\min\frac{1}{2}\sum_{i=1}^{n}\rho \left({\left\|p_{i}^{c}-Tp_{i}^{w}\right\|}^{2}\right)$$
where *ρ*(·) is the robust kernel function and *T* is the robot pose. 

If the number of optimized inner points is greater than 50, the current frame is judged to be successfully relocated. If the number of candidate keyframes is too small, it may not be possible to find a frame that is sufficiently similar to the current frame, or if the similarity between the found frame and the current frame is too low, then using image retrieval methods may not be able to successfully recover the robot pose. Therefore, in this case, the “image–map matching” method is used for relocation. It is important to note that the approach estimates the robot pose by directly matching the feature points of the current frame with the map points, which does not rely on the number of candidate keyframes.

When the number of candidates keyframes *N_cand_* < 2, it becomes difficult and unreliable to use image matching for relocation. So, the “image–map matching” relocation method is used in this case. First, the local map is updated, i.e., for each visual word *v_ci_* in the bag-of-words vector *v_c_* of the current frame and, if its TF-IDF [31] value in the current frame is greater than the threshold *v*th, it can be employed to characterize the current frame, and the index of the corresponding leaf node of the visual word in the K-tree is used to quickly find all map points that contain the visual word to obtain a subset *M* = {*m_1_*,…, *m_n_*} of map points. Then, the nearest neighbor search algorithm [32] is used to fast match the feature point *p_i_* of the current frame with the map point *m_i_* in the subset of map points. On this basis, using (*p_i_*, *m_i_*) to represent the matching relationship between feature points and map points, the set of matching point pairs between feature points and map points is {(*p_i_*, *m_i_*)${\}}_{i=1}^{n}$. During the matching process, the EPnP algorithm is utilized to improve the efficiency and accuracy, and then the robot’s pose *T* is solved based on the obtained 2D–3D matching points, i.e.,
(25)$$T=\arg\min\frac{1}{2}\sum_{i=1}^{n}\rho \left({\left\|p_{i}-\pi \left(Tm_{i}\right)\right\|}^{2}\right)$$
where *π* is the projection function and *ρ*(·) is the robust kernel function. Using the Levenberg–Marquadt method to solve the above equation can effectively improve the accuracy of the pose solution and finally successfully recover the current camera tracking.

The image–map matching relocation method can effectively improve the success rate of relocation, but it may cause drift after restoring the mobile robot’s pose, so loop closure detection can be adopted to eliminate the accumulated drift of the system. When the system detects a closed loop, the global BA is performed to continue to optimize the mobile robot’s pose. As the robot continues to move, the number of keyframes and map points in the system will continue to increase, which will place a huge burden on the system optimization. The location of map points in the scene will converge to a value after being observed many times, and it is often futile to continue optimizing the location of map points. Therefore, the system is optimized only for certain moments of the mobile robot poses, i.e., the pose graph is used for optimization, and the pose points of the optimized map are robot poses and map points, and the edges represent the constraint relations between them. Assuming the pose of the mobile robot at each pose point is *T*_1_, *T*_2_, *T*_3_,…, *T_n_*, then, the motion relationship between any two *T_i_* and *T_j_* can be expressed as:(26)$$T_{ij}=T_{i}^{-1}T_{j}$$

In terms of BA optimization, the error is established as follows:(27)$$e_{ij}=\ln\left(T_{ij}T_{i}^{-1}T_{j}\right)$$

Further, the global optimization is performed for all position points of the robot using the following overall objective function: (28)$${\left(T_{1},T_{2},\cdots ,T_{n}\right)}_{opt}=\min\frac{1}{2}\sum_{i,j\in \xi }e_{ij}^{T}\sum_{ij}^{-1}e_{ij}$$
where *ξ* is all edges in the pose graph. 

The use of the global BA algorithm can further optimize the mobile robot poses and map points, reduce the cumulative drift of the whole system, and improve the accuracy and robustness of the system.

### 3.6. 3D Navigation Map Building

In our work, to enable the map to be used for robot navigation and autonomous obstacle avoidance, we capitalize on the RGB image information, depth value, and keyframes’ pose to build a 3D dense point cloud map of the scene, and then combine the octree map to further generate a 3D navigation dense map. Typically, 2D points can be converted into 3D point clouds by using the RGB information and depth value of the keyframes, and the conversion relationship is as follows:(29)$$s\left[\begin{array}{l} u\\ v\\ 1 \end{array}\right]=K\left(R|t\right)\left[\begin{array}{c} x_{w}\\ y_{w}\\ z_{w} \end{array}\right]$$
where *s* is the scaling factor of depth value, *K* is the camera internal parameter, *R* and *t* are the rotation matrix and translation vector, respectively, [*u v* 1]*^T^* are the pixel point coordinates, and [*x_w_ y_w_ z_w_*] are the 3D points in space. According to Equation (29), the coordinates of the real point cloud can be obtained as follows:(30)$$\left[\begin{array}{c} x_{w}\\ y_{w}\\ z_{w} \end{array}\right]=\left[\begin{array}{l} \frac{z\left(u-c_{x}\right)}{f_{x}}\\ \frac{z\left(u-c_{y}\right)}{\begin{array}{l} f_{y}\\ z \end{array}} \end{array}\right]$$
where *c_x_*, *c_y_*, *f_x_*, *f_y_* denote the camera internal parameters. 

When constructing a dense map, it is susceptible to the influence of point clouds generated by dynamic objects. If the map contains information about dynamic objects, the robot cannot determine whether the object exists during navigation or autonomous obstacle avoidance. Hence, in our case, the dynamic points are judged based on the optical flow amplitude of each pixel in the image. During the process of map building, once a keyframe is inserted, each pixel in the keyframe needs to be traversed. First, these pixels with abnormal depth values are eliminated, and then the magnitude of the optical flow amplitude of the pixel point is determined. If its optical flow amplitude is greater than the threshold, the point is determined as a dynamic point and removed from the dense point cloud. After eliminating the dynamic point cloud, the static point cloud is compensated by keyframes from different perspectives to construct a complete static dense point cloud map. Due to the existence of certain redundant information between adjacent keyframes, a large number of repeated point clouds will be generated in the process of constructing a dense map, resulting in an increase in required storage space. Therefore, voxel filtering is employed to down-sample point clouds, which not merely maintains the shape features of the point cloud, but removes certain noise, smooths the interval between point clouds, and finally adopts statistical filtering to filter out outliers again. The process of 3D dense map construction is shown in Figure 6.

Although the dense point cloud map can intuitively represent each object in the scene, the point cloud map occupies a large amount of storage space and cannot be used for robot navigation and obstacle avoidance. Therefore, this paper further constructs an octree map based on the dense point cloud map. The leaf nodes of the octree map represent the occupancy state of each voxel in the form of probability. If a leaf node is observed as being occupied, the probability is constantly approaching 1. If the leaf node is observed as unoccupied, the probability is constantly approaching 0. To ensure that the probability does not exceed the interval, the logit function is usually used instead of the probability description, i.e.,
(31)$$l=\log it\left(p\right)=\log\frac{p}{1-p}$$
where *l* is the logarithm of probability, and *p* is the probability of 0~1. When a node is continuously observed to be occupied, *l* will continue to increase; when the node is unoccupied, *l* will continue to decrease. Assuming that *n* denotes a node and the observed value is *z*, the probability logarithm of the node from the beginning to time *t* is *L*(*n*|*z*_1:*t*_), then, at time *t* + 1, the probability logarithm is established as follows: (32)$$L\left(n|z_{1:t+1}\right)=L\left(n|z_{1:t}\right)+L\left(n|z_{t+1}\right)$$

Apparently, adopting this form can make map updates more flexible and convenient.

## 4. Simulations and Experiments

To verify the effectiveness of the proposed algorithm, a series of simulations and experiments were conducted on the TUM public dataset and real scenes, respectively. The experimental platform was as follows: Intel Core i5-7300HQ, 2.4 GHz CPU, with 16 G RAM, no GPU acceleration, running under the Ubuntu 18.04 operating system.

### 4.1. Simulation Analysis with TUM Dataset

In this work, the “walking” and “sitting” dynamic sequences from the TUM RGB-D datasets [33] were selected to test the performance of the SLAM algorithm in dynamic scenes. The dynamic sequences correspond to different camera motion modes and scene structures, including static, xyz, halfsphere, and rpy. Each sequence contains RGB images, depth images, and the system’s true value trajectory, which can be used to evaluate the system’s localization accuracy. Pedestrians walked around a desk in the dynamic sequence, while the camera was in constant motion, putting forward high requirements for the robustness of the SLAM system. 

To validate the capability of dynamic point culling in the front-end of the proposed algorithm, experiments were conducted on two scenarios: the camera, relative to the background, being stationary and moving. Figure 7 displays the running results of ORB- SLAM2 and our algorithm under the walking_xyz sequence, where the 781st frame was a motion scene with the camera stationary relative to the background. It can be found that ORB-SLAM2 extracted many features of pedestrians, while our algorithm successfully obtained the dynamic regions in the scene by the motion detection algorithm and rejected the features falling in the dynamic regions. Figure 8 illustrates the running effects of the two tested algorithms under the walking_half_sphere sequence, where the 58th frame was a scene of camera motion relative to the background. It is clear that, compared to the ORB-SLAM2, our algorithm effectively removed feature points in dynamic regions. Therefore, it can be concluded that the front-end of our algorithm can successfully detect the dynamic regions and reject their feature points in both cases.

To evaluate the error performance of the algorithm in dynamic scenes, a series of simulations were conducted on the highly dynamic “walking” sequence and slightly dynamic “sitting” sequence of the TUM datasets, and then absolute trajectory error (ATE) and relative pose error (RPE) were employed as indicators for system accuracy. Herein, ATE was used to evaluate the accuracy of SLAM systems, which directly calculates the error between the actual and estimated poses of the system, reflecting the global positioning accuracy of the system; RPE calculates the difference between the actual and estimated pose changes in the same time interval to evaluate the drift of the SLAM system, including relative translation error (RTE) and relative rotation error (RRE). In this study, to fully demonstrate the effectiveness of the proposed algorithm in dynamic scenes, in addition to comparing with ORB-SLAM2 [34], the other five dynamic SLAM algorithms were selected, including DVO + MR [35], BAMVO [36], DynaSLAM, and two different methods proposed by Dai et al. [16] and Sun et al. [11], respectively. 

The RMSE comparisons of RTE and RRE between our algorithm and other methods under different image sequences are respectively reported in Table 1 and Table 2. Apparently, our algorithm achieved better rotation and translation RMSEs under most of the image sequences. In slightly dynamic sequences, the performance of our algorithm was comparable to other algorithms. Since the ORB-SLAM2 directly incorporates the feature points of dynamic objects into pose optimization without processing them, its pose estimation drift was significant, and the RMSEs of relative rotation and relative translation were greater than other methods. For all highly dynamic image sequences, the RTE and RRE of our algorithm outperformed the DVO + MR and BAMVO algorithms, with an average reduction of 79.6% and 80.4% in translation RMSE and 78.9% and 87.1% in rotation RMSE. Remarkably, due to the large rotation and fast motion speed of the camera along the hemispherical trajectory in the walking_rpy sequence, the dynamic threshold of the proposed algorithm changes significantly, and the detection effect of dynamic objects was poor. Thus, in the walking_rpy sequence, the translation and rotation RMSEs of our algorithm were higher than that of the dense optical flow method, but in the other three highly dynamic sequences, the rotation and translation errors of our algorithm were better than that of the dense optical flow method. Compared with the geometric method, our algorithm performed better in walking_static and walking_xyz sequences, and in the walking_half and walking_rpy sequences, the positional errors of our algorithm and the geometric method were on the same order of magnitude. From Table 1 and Table 2, it can also be found that our algorithm performed poorly compared to deep learning-based DynaSLAM on walking sequences, but on the other three sequences, the error in our algorithm was essentially the same as the one in the DynaSLAM.

Table 3 compares the RMSE for the ATE among different algorithms under three types of image sequences. It is clear that the error of our method was basically on the same order of magnitude as other algorithms in both static and slightly dynamic scenes. In highly dynamic scenes, although the ORB-SLAM2 optimized the camera pose by back-end projection, because of too many feature points generated by dynamic objects, the ATE of ORB-SLAM2 was relatively large. Instead, our algorithm detected and eliminated dynamic points in the front-end, resulting in a 92.7% improvement in accuracy compared to ORB-SLAM2. From Table 3, it can be seen that the two deep learning-based methods DynaSLAM and DS-SLAM had higher accuracy. The accuracy of our method was not as good as DynaSLAM, but it was similar to DS-SLAM, and it is worth mentioned that our method did not involve a GPU. Additionally, compared with the other three tested algorithms, our algorithm performed well in walking_static, walking_xyz and walking_half sequences, but the dynamic detection result of our algorithm was poor due to the excessive camera motion in walking_rpy sequences, so the localization accuracy of our algorithm in walking_rpy sequences was relatively low.

Figure 9 plots the ATE comparison between the proposed algorithm and ORB-SLAM2 under the walking_xyz sequence and the walking_half sequence, respectively. The black curve represents the real trajectory of the camera, the blue curve represents the estimated trajectory of the algorithm, and the red region represents the error between the two. It can be seen that the ATE of our algorithm in the two sequences was better than that of ORB-SLAM2, indicating that our algorithm has high accuracy in dynamic scenes. Figure 10 compares the RPE between our algorithm and ORB-SLAM2 in walking_xyz and walking_half sequences. It can be seen that the trajectory drift of our algorithm is smaller, and it performs better in the face of continuous camera pose transformation in dynamic scenes.

Figure 11 compares the number of keyframes in different image sequences between our algorithm and ORB-SLAM2. In the fr1_room sequence, the camera moves continuously and rapidly, and there is a large amount of rotational motion. Since our method selects keyframes based on the size of the camera pose transformation, the number of keyframes extracted by our algorithm was greater than that of ORB-SLAM2 under the fr1_room sequence. In highly dynamic sequences, e.g., fr3_walking_xyz, fr3_ walking_half, fr3_walking_static, and fr3_walking_rpy, due to the interference of dynamic feature points, the number of inner points of ORB-SLAM2 was relatively small, so the tracking was maintained by adding a large number of keyframes. However, our algorithm used static feature points to estimate camera pose, which can not only maintain stable tracking but also eliminate a large number of low-quality keyframes by judging the dynamic characteristics of the current frame. It reveals that the number of keyframes of our algorithm in highly dynamic datasets was much smaller than that of ORB-SLAM2, reducing the redundancy of keyframes and improving the optimization efficiency of the system in dynamic scenes.

Generally, the mean tracking time of tracking threads and mean sampling frequency could reflect the real-time performance and running efficiency of SLAM systems. In our work, the front-end of the system achieves fast tracking through the semi-direct method. Although the method exploited dense optical flow to complete dynamic detection, the computational complexity of optical flow was reduced by performing down-sampling on the image, so the whole system still has high running efficiency. The comparison results of running efficiency for our algorithm, ORB-SLAM2, DS-SLAM, and the algorithm proposed in [37] under the fr3_walking_xyz sequence are reported in Table 4. It is clear that the mean tracking time for our algorithm to successfully track a frame was 45.576 ms, so our algorithm had the greatest increase in time consumption with respect to ORB-SLAM2 compared to other tested algorithms. Meanwhile, compared with the other two algorithms involving dynamic detection, our algorithm did not merely take the least time but also ran at a higher sampling frequency of 21.941 Hz, which implied that it has better real-time performance and running efficiency. Moreover, we also list the simulation results of our method without using down-sampling processing in Table 4; this method in Table 4 is denoted as “Ours *”. Apparently, the method in this paper adopted down-sampling with faster tracking speed and higher sampling frequency. Therefore, the comparison results demonstrate that our algorithm successfully reduced the computational complexity of the system and improved running efficiency.

In dynamic scenes, after the interference of dynamic feature points, there are often situations where the number of matching points is too small and there are a large number of mismatches in the matching points, which leads to pose tracking failure and is not conducive to the long-term autonomous navigation of mobile robots. To verify the robustness of the algorithm in dynamic scenes, the proposed algorithm and ORB-SLAM2 were run five times under highly dynamic sequences walking_xyz, walking_half and walking_rpy and their tracking loss count and successful recovery pose count were calculated. From Table 5, it can be seen that ORB-SLAM2 experienced pose tracking loss in all three highly dynamic sequences, while the proposed algorithm tracked stably in the walking_xyz and walking_half sequences, without any pose loss. It only lost the pose once in the walking_rpy sequence but successfully retrieved the pose through relocation. Therefore, it can be concluded that our algorithm has stronger robustness in highly dynamic scenes.

### 4.2. A Case Study with Mobile Robot in Real Scene

To verify the feasibility and effectiveness of the proposed algorithm in real dynamic scenes, a case study was conducted in an indoor dynamic scene utilizing a mobile robot equipped with an ASUS Xtion depth camera. The depth camera was composed of three main components: an RGB camera, an infrared structured light emitter, and a receiver. Its parameters are listed in Table 6. The experimental scene was a laboratory with an area of 8.4 m × 6.4 m, as shown in Figure 12. A reference trajectory with a length of 6 m and a width of 1 m was set in the scene, and a pedestrian constantly walked back and forth to simulate a real dynamic environment.

To evaluate the capability of eliminating dynamic points in the front-end of our algorithm, a comparative experiment was conducted between the proposed algorithm and ORB-SLAM2 in a real dynamic scene. During the experiment, the mobile robot moved forward at a constant speed along the reference trajectory, and a pedestrian walked in front of the robot to simulate the dynamic object. Figure 13 and Figure 14 demonstrate the feature point extraction results and sparse pose graph of the two tested algorithms. We could reasonably conclude from Figure 13a–d that the ORB-SLAM2 extracted a certain number of feature points of pedestrians. Thus, from Figure 13f,g, we find that the trajectory of ORB-SLAM2 in sparse maps had undergone significant drift to the right. It is immediately apparent from Figure 14a–d that our algorithm adopted dense optical flow to detect and remove dynamic features, so no dynamic feature points were extracted from pedestrians. It is further seen from Figure 14e–h that, due to not being affected by dynamic feature points, the trajectories in the sparse map constructed by our algorithm did not drift. This indicated that our algorithm can effectively detect dynamic regions and remove dynamic features, thereby improving the localization accuracy of the algorithm in dynamic scenes. 

To further verify the effectiveness of the proposed relocation method and the global localization accuracy in real dynamic scenes, comparison experiments were conducted between our algorithm and ORB-SLAM2 in a laboratory environment and manually interrupted for relocation. Figure 15 compares the trajectories estimated by our algorithm and ORB-SLAM2 in a real dynamic scene, where point A was the location of manual interruption (i.e., intentionally obstructing the camera’s field of view). We observed that, in the case of manual interruption and loss of pose, both our algorithm and ORB-SLAM2 can successfully retrieve the pose. However, with the presence of a large number of dynamic points in the map of ORB-SLAM2, there was still significant trajectory drift after the pose was successfully restored by feature matching. In contrast, for the entire process, our algorithm only matched with keyframes and static map points to restore the pose by eliminating dynamic points. Although the robot’s pose might include some errors after successful relocation, the overall drift degree was relatively small. In terms of global trajectory, ORB-SLAM2 introduces a large number of dynamic points in pose optimization, resulting in a large drift of its overall trajectory compared with the reference trajectory. In the vicinity of point B, on account of the slow moving speed of the robot and the long occupation time of pedestrians in the scene, some dynamic points were added to the pose estimation, so our algorithm produces a small trajectory drift, while the rest were not significantly affected by dynamic objects, and the overall trajectory of our algorithm basically fitted with the reference trajectory. Consequently, the experimental results show that our algorithm had high localization accuracy and robustness in a dynamic scene. 

Figure 16 illustrates the 3D dense local map and the corresponding octree map constructed by different algorithms in dynamic scenes. Clearly, the dense point cloud map shown in Figure 16a did not exclude dynamic objects, so a large number of dynamic point clouds can be observed in the red box of the map, and voxel blocks containing a large number of dynamic objects were found in Figure 16c. As compared to the ORB-SLAM2, most dynamic point clouds have been removed in the dense map shown in Figure 16b, and only static object point clouds were included. The corresponding scene in Figure 16d only contains a small number of voxel blocks of dynamic objects. Similar results can be found from the global map in Figure 17. Although ORB-SLAM2 can complete the final map building, it contained clear moving objects and deviations (Figure 17a,b), whereas our method generated a consistent map that only included a few residual features of moving objects, and there was a slight overlap and distortion in the map at the manual interruption point A. Therefore, these results clearly indicate that the proposed algorithm does not merely effectively eliminate the dynamic point clouds but generates a consistent and accurate 3D navigation map, thereby demonstrating the feasibility and effectiveness of our method.

## 5. Conclusions and Future Works

This paper proposed a robust semi-direct 3D SLAM algorithm for mobile robots based on dense optical flow in dynamic scenes. The front-end of the algorithm adopted the semi-direct method to calculate the robot’s pose, compensated the image through the homography matrix, and combined with the dynamic threshold to separate the dynamic region in the scene. On this basis, the feature points falling in the dynamic region were eliminated, and the static feature points were used to estimate the robot’s motion, which improved the accuracy and robustness of the algorithm in the dynamic scene. Meanwhile, considering the importance of keyframes to system optimization and mapping, a high- performance keyframe selection strategy was constructed by combining the pose transformation and the dynamic degree of the scene, which improves the quality of keyframes and reduces the complexity of back-end optimization. Furthermore, we also developed a relocation method that combines keyframes and map points, solving the problem of difficulty in retrieving the robot pose after direct tracking failure. Specifically, feature extraction was performed on the current frame that had failed tracking, and the number of candidate keyframes was used to determine whether to use image matching or map matching to restore pose, thereby improving the success rate of relocation. Finally, the effectiveness of the algorithm was verified through a series of simulations and experiments.

Considering the dense optical flow has a large calculation error in the case of light changes or noise interference, it will lead to inaccurate tracking and detection of dynamic objects. In future work, we plan to employ a more stable optical flow network to calculate dense optical flow, which can more accurately distinguish the dynamic objects and static backgrounds, thereby reducing false positives and missed detections. Another perspective of this research is to combine a lightweight segmentation network or object detection network to constrain the calculation of optical flow, so as to more accurately detect and track dynamic objects. Indeed, the method further reduces the impact of dynamic objects on pose estimation, while ensuring the real-time performance of the algorithm. As our algorithm only considers the photometric error of the image during the initial estimation stage of the robot pose, the depth image obtained by the RGB-D camera can provide accurate distance information of the scene. Therefore, we also plan to introduce the depth error of pixel points into pose estimation to more accurately estimate the robot’s pose. In addition, there are often different lighting transformations in actual scenes, which have a significant impact on semi-direct methods and dense optical flow calculations. Although inter-frame brightness compensation can reduce the impact of some lighting changes, the algorithm still performs poorly in some extreme cases. As part of future work, we intend to introduce the photometric imaging model into the semi-direct method, reduce the impact of lighting changes on the algorithm by performing photometric correction on the image, and consider combining other types of sensors with the RGB-D camera for data fusion, thereby further improving the robustness of the algorithm and expanding its application scenes.

## Figures and Tables

**Figure 1 biomimetics-08-00371-f001:**
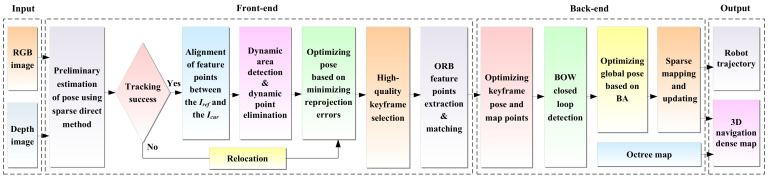
Scheme of the robust semi-direct 3D SLAM for mobile robot based on dense optical flow in dynamic scenes.

**Figure 2 biomimetics-08-00371-f002:**
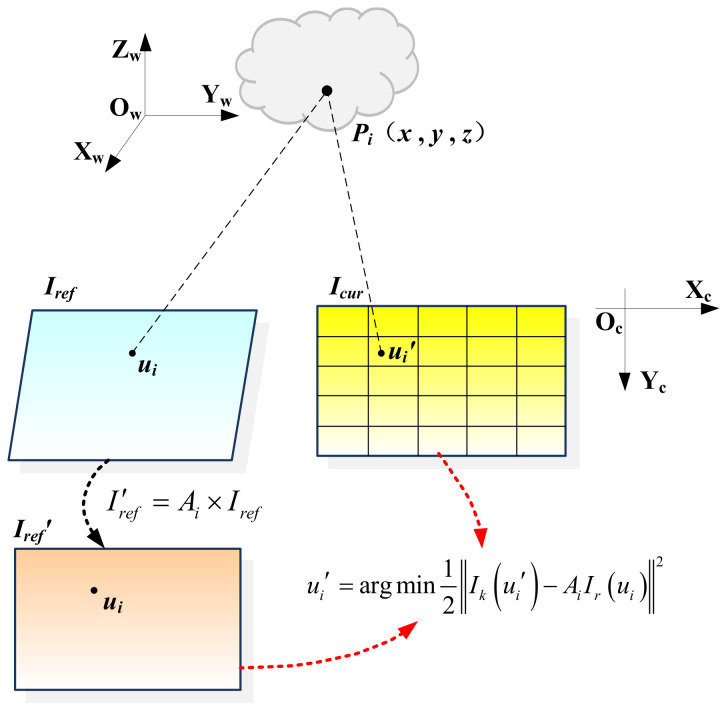
Calculation of correspondence between pixel points.

**Figure 3 biomimetics-08-00371-f003:**
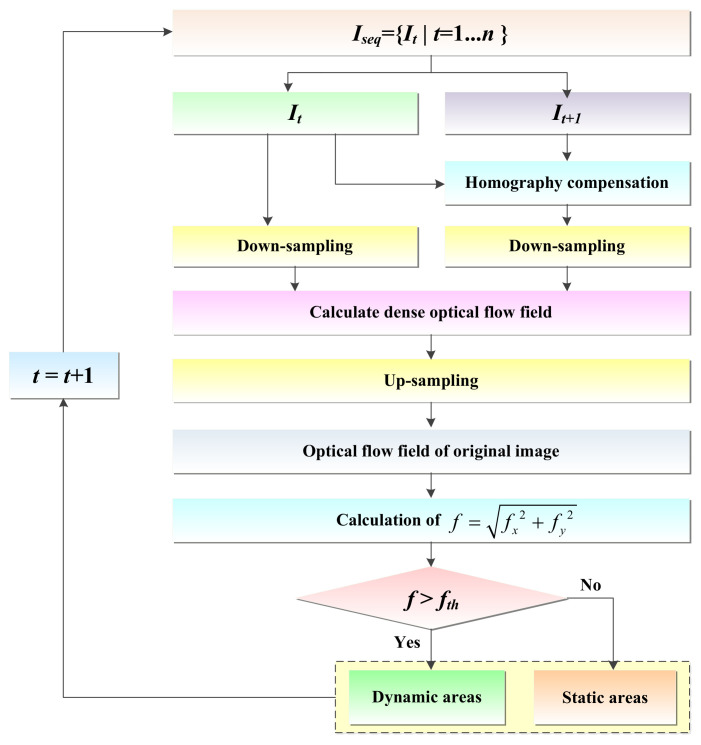
Scenarios for the dynamic areas’ detection.

**Figure 4 biomimetics-08-00371-f004:**
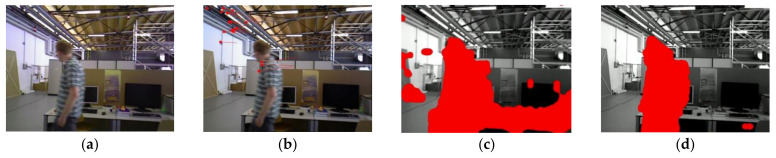
Comparison of the results for different optical flow detection methods: (**a**) original image; (**b**) sparse optical flow; (**c**) dense optical flow; (**d**) dense optical flow with homography compensation.

**Figure 5 biomimetics-08-00371-f005:**
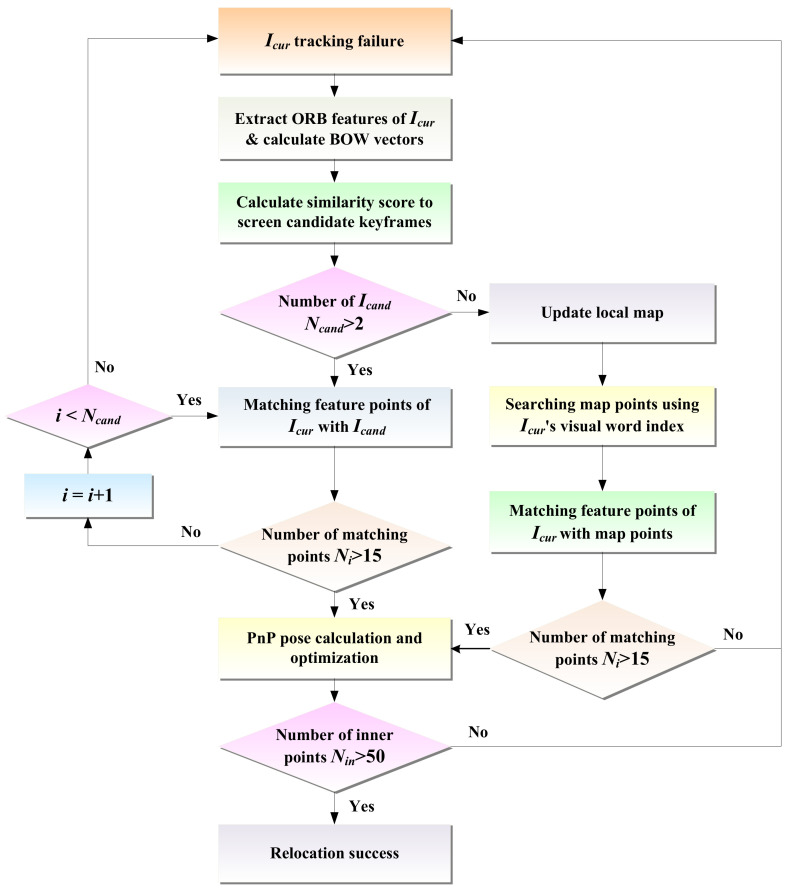
Scenarios for the relocation process.

**Figure 6 biomimetics-08-00371-f006:**
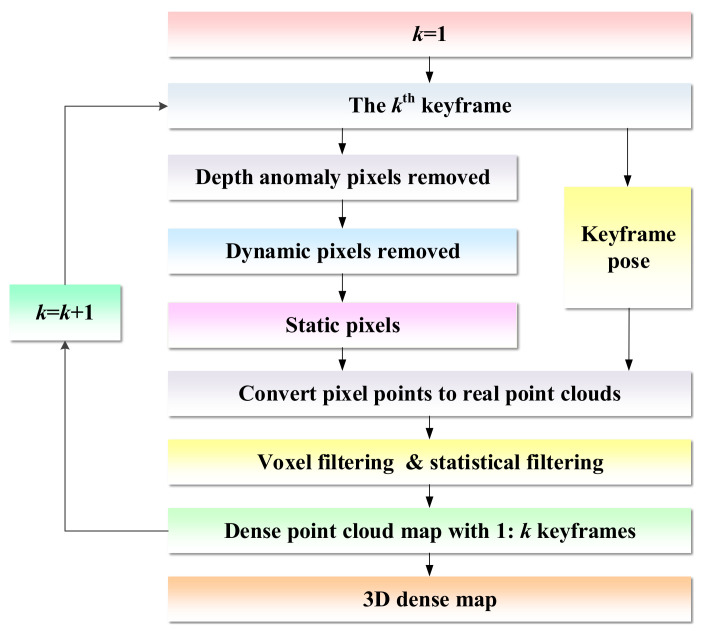
Scenarios for the 3D dense map construction.

**Figure 7 biomimetics-08-00371-f007:**
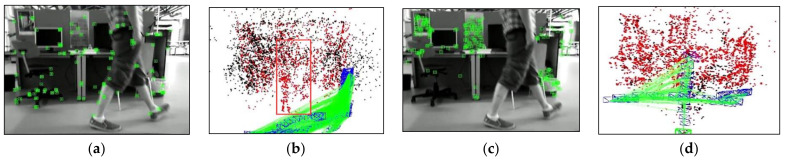
Comparison of the dynamic point rejection effects in the case of the camera being stationary, (red box represent dynamic point clouds): (**a**,**b**) are feature point extraction results and corresponding sparse pose graph of ORB-SLAM2, respectively; (**c**,**d**) are feature point extraction results and corresponding sparse pose graph of ours, respectively.

**Figure 8 biomimetics-08-00371-f008:**
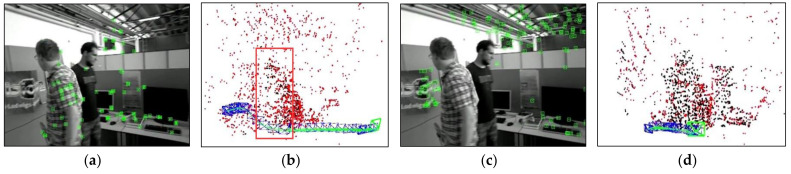
Comparison of dynamic point rejection effect in the case of the camera being in motion(red box represent dynamic point clouds): (**a**,**b**) are feature point extraction results and corresponding sparse pose graph of ORB-SLAM2, respectively; (**c**,**d**) are feature point extraction results and corresponding sparse pose graph of ours, respectively.

**Figure 9 biomimetics-08-00371-f009:**
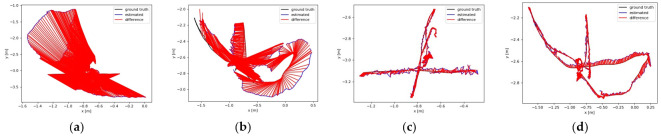
Comparison of the absolute trajectory error (ATE) under different image sequences: (**a**,**b**) are the results of ORB-SLAM2 under walking_xyz and walking_half sequences, respectively; (**c**,**d**) are the results of our method under walking_xyz and walking_half sequences, respectively.

**Figure 10 biomimetics-08-00371-f010:**
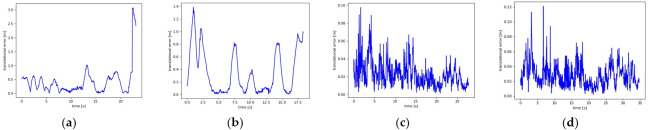
Comparison of the relative pose errors (RPEs) under different image sequences: (**a**,**b**) are the results of ORB-SLAM2 under walking_xyz and walking_half sequences, respectively; (**c**,**d**) are the results of our method under walking_xyz and walking_half sequences, respectively.

**Figure 11 biomimetics-08-00371-f011:**
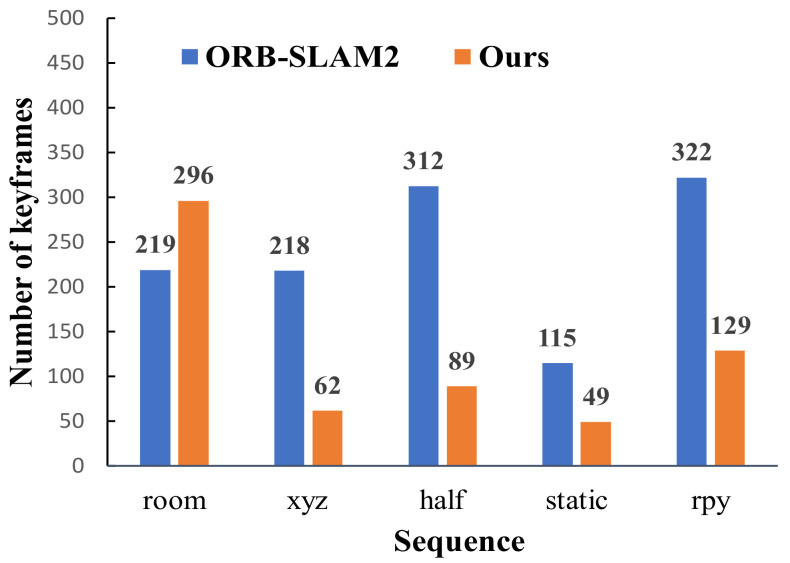
Comparison of the number of keyframes under different image sequences.

**Figure 12 biomimetics-08-00371-f012:**
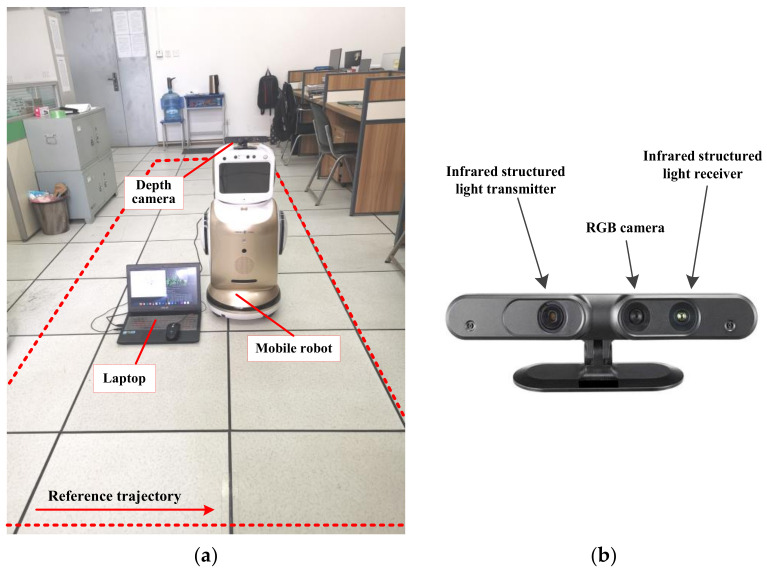
Experimental scene: (**a**) the experimental platform of the mobile robot and the reference trajectory; (**b**) the ASUS Xtion depth camera used by the mobile robot.

**Figure 13 biomimetics-08-00371-f013:**
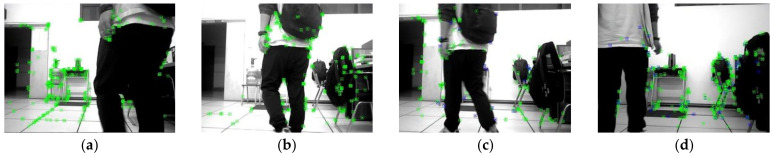
Feature point extraction results and corresponding sparse pose graph of ORB-SLAM2 in real dynamic scene. (**a**–**d**) are the feature point extraction results of ORB-SLAM2, (**e**–**h**) are the sparse pose graph of ORB-SLAM2.

**Figure 14 biomimetics-08-00371-f014:**
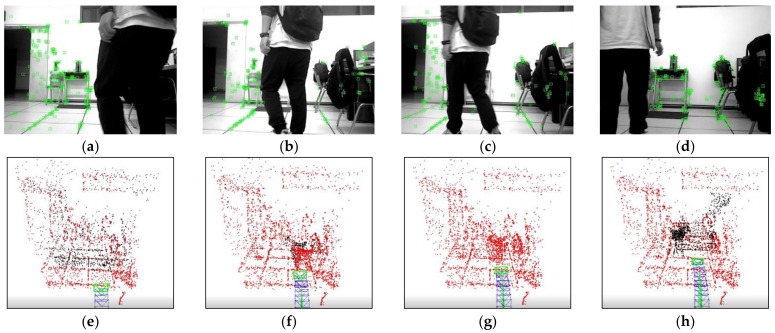
Feature point extraction results and corresponding sparse pose graph of the proposed algorithm in real dynamic scene. (**a**–**d**) are the feature point extraction results of ours, (**e**–**h**) are the sparse pose graph of ours.

**Figure 15 biomimetics-08-00371-f015:**
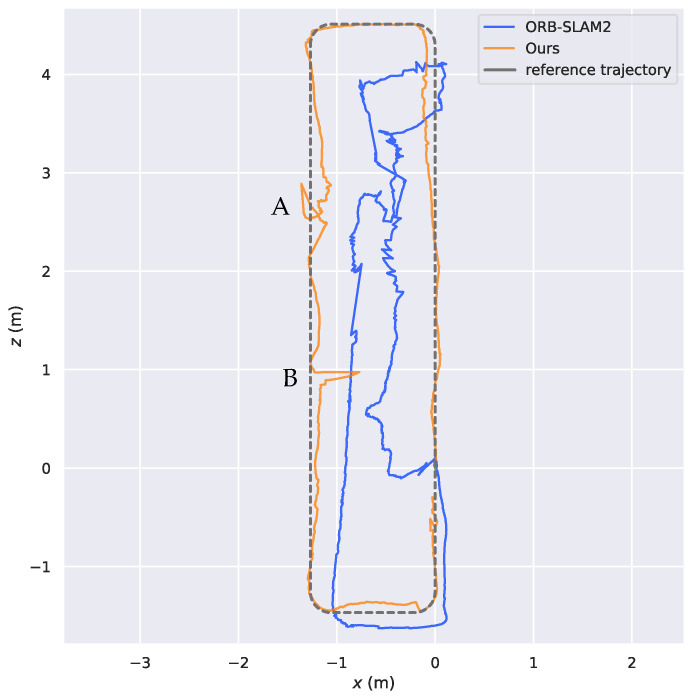
Comparison of the trajectories estimated by different algorithms in dynamic scene.

**Figure 16 biomimetics-08-00371-f016:**
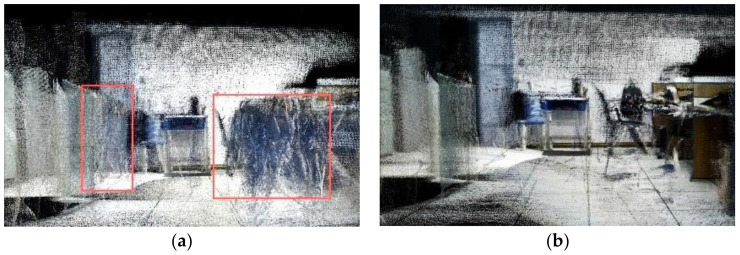
Comparison of the 3D dense local map and the corresponding octree map construction by different algorithms in dynamic scene (red box represent dynamic point clouds): (**a**,**b**) are dense maps of ORB-SLAM2 and ours, respectively; (**c**,**d**) are octree maps of ORB-SLAM2 and ours, respectively.

**Figure 17 biomimetics-08-00371-f017:**
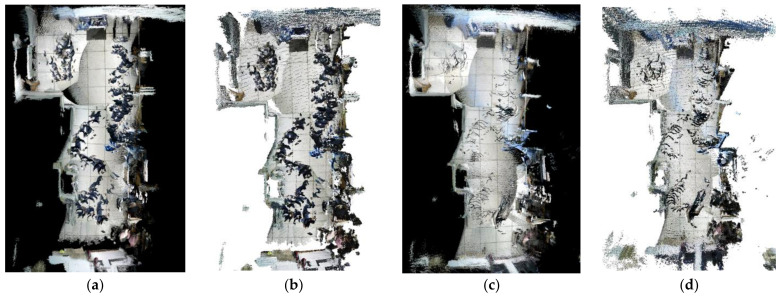
Comparison of the 3D dense global map and the corresponding octree map construction by different algorithms in dynamic scene: (**a**,**b**) are dense map and the corresponding octree map of ORB-SLAM2; (**c**,**d**) are dense map and the corresponding octree map of ours.

**Table 1 biomimetics-08-00371-t001:** Comparison of RMSE for relative translation error (RTE) among different algorithms.

Sequences		Translation RMSE/(m/s)						
		ORB-SLAM2	DVO + MR	Geometric	Dense Optical Flow	BAMVO	DynaSLAM	Ours
Static Environment	fr2_desk	0.1279	-	-	-	0.0299	-	0.1217
	fr3_long_house	0.0123	-	-	-	0.0332	-	0.0125
Slightly Dynamic Environment	fr2_desk_person	0.0110	0.0172	0.0362	0.0213	0.0352	-	0.0112
	fr3_sitting_static	0.0158	-	0.0138	-	0.0248	0.0126	0.0145
	fr3_sitting_xyz	0.0134	0.0330	0.0134	0.0357	0.0482	0.0208	0.0153
	fr3_sitting_rpy	0.0386	-	0.0320	-	0.1872	-	0.0437
Highly Dynamic Environment	fr3_walking_static	0.1557	0.0842	0.0141	0.0307	0.1339	0.0133	0.0076
	fr3_walking_xyz	1.0913	0.1214	0.1266	0.0668	0.2326	0.0254	0.0269
	fr3_walking_half	1.0289	0.1672	0.0517	0.0611	0.1738	0.0394	0.0317
	fr3_walking_rpy	1.2863	0.1751	0.2299	0.0968	0.3584	0.0415	0.2123

**Table 2 biomimetics-08-00371-t002:** Comparison of RMSE for relative rotation error (RRE) among different algorithms.

Sequences		Rotation RMSE/(°/s)						
		ORB-SLAM2	DVO + MR	Geometric	Dense Optical Flow	BAMVO	DynaSLAM	Ours
Static Environment	fr2_desk	1.8524	-	-	-	1.1167	-	1.6889
	fr3_long_house	0.0249	-	-	-	2.1583	-	0.0257
Low Dynamic Environment	fr2_desk_person	0.4491	0.7341	1.3951	0.7744	1.2159	-	0.4582
	fr3_sitting_static	0.3633	-	0.3786	-	0.6997	0.3416	0.3775
	fr3_sitting_xyz	0.5817	0.9828	0.5729	1.0362	1.3885	0.6249	0.5982
	fr3_sitting_rpy	0.9358	-	0.9047	-	5.9834	-	1.2253
High Dynamic Environment	fr3_walking_static	7.0757	2.0487	0.3293	0.8998	2.0833	0.3000	0.2623
	fr3_walking_xyz	18.6722	3.2346	2.7413	1.5950	4.3911	0.6252	0.6530
	fr3_walking_half	23.1243	4.3755	0.9854	1.9004	4.2863	0.8933	0.8729
	fr3_walking_rpy	21.6317	5.0108	4.6327	2.5936	6.3398	0.9047	4.1323

**Table 3 biomimetics-08-00371-t003:** Comparison of the RMSE for absolute trajectory error (ATE) among different algorithms.

Sequences		ATE						
		ORB- SLAM2	DVO + MR	Geometric	Dense Optical Flow	DynaSLAM	DS-SLAM	Ours
Static Environment	fr2_desk	0.0841	-	-	-	-	-	0.0809
Slightly Dynamic Environment	fr2_desk_person	0.0739	0.0596	0.0075	0.0759	-	-	0.0768
	fr3_sitting_static	0.0847	-	-	-	0.0108	0.0065	0.0801
	fr3_sitting_xyz	0.0090	0.0482	0.0091	0.0514	0.0159	-	0.0122
	fr3_sitting_rpy	0.0313	-	0.0025	-	-	-	0.0385
Highly Dynamic Environment	fr3_waling_static	0.4231	0.0656	0.0108	0.0334	0.0069	0.0082	0.0080
	fr3_walking_xyz	0.6903	0.0932	0.0874	0.0657	0.0155	0.0246	0.0227
	fr3_walking_half	0.4325	0.1252	0.0354	0.0668	0.0257	0.0311	0.0289
	fr3_walking_rpy	0.9527	0.1333	0.1608	0.0729	0.0378	0.4438	0.3328

**Table 4 biomimetics-08-00371-t004:** Comparison of running efficiency among different algorithms.

Algorithms	Mean Tracking Time/ms	Mean Sampling Frequency/Hz
ORB-SLAM2	25.979	38.492
Sparse optical flow	53.455	18.707
DS-SLAM	85.257	11.729
Ours *	69.332	14.423
Ours	45.576	21.941

* represents our method without using down-sampling processing.

**Table 5 biomimetics-08-00371-t005:** Comparison of tracking loss and relocation times of different algorithms.

Sequence	Number of Lost Tracks		Number of Relocations	
	ORB-SLAM2	Ours	ORB-SLAM2	Ours
walking_xyz	4	0	3	0
walking_half	5	0	2	0
walking_rpy	3	1	2	1

**Table 6 biomimetics-08-00371-t006:** Detailed parameters of the ASUS Xtion depth camera.

Performance Indicators	Parameter Values
Size of depth camera	18 cm × 3.5 cm × 5 cm
Effective measurement range	0.8~3.5 m
Field of view angle	58° H × 45° V × 70° D
Sampling frame rate	30 frames/s
Size of image obtained	680 × 480

## Data Availability

Not applicable.

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
