# Peer review of "A Robust Semi-Direct 3D SLAM for Mobile Robot Based on Dense Optical Flow in Dynamic Scenes"

_biomimetics, 2023, doi:10.3390/biomimetics8040371_

Round 1

Reviewer 1 Report

This paper investigates a method for Mobile Robot’s SLAM based on dense optical flow. This is not new but still an interesting topic. Generally, the paper is technically sound and well organized. The method is described and validated through simulations and experimental case studies. However, there is still room for improvement.

1.     As mentioned, this topic is not new and has been investigated in many existing papers. It is seen that Lidar is also a good and popular sensor that can provide effective solutions and deliver the preferred accuracy for the 3D SLAM problem. Many related studies have been published in recent years. The authors may want to compare their work with some existing (recently published) Lidar-based work to highlight the contributions of this paper.

2.     The term “simulation experiment” is quite strange.

3.     In Figure 15, a reference trajectory should be shown so that readers can evaluate the performance of the algorithms.

Minor editing is recommended. 

Reviewer 2 Report

Interesting study on robust semi-direct 3D SLAM algorithm based on dense optical flow relying on initial pose estimation of robots and detecting the dynamic region of image sequences by dense optical flow. It is obvious to  consider tracking failure in dynamic scenes with the help of ORB feature points and matching them with keyframes  or map points to recover the robot pose. Your global optimization approach applied the the robot’s pose and to sparse map points is plausible. The use of a 3D dense octree map supporting robot navigation and obstacle avoidance is reasonable. The proposed high-precision dynamic object detection method based on dense optical flow capable to determine dynamic regions of images is plausible.  The keyframe selection strategy reducing the influence of dynamic objects on the quality of keyframes improves obviously the accuracy of the  algorithm.  Your relocation method relying on feature point  extraction and matching shows a good success rate in terms   robustness. The presented overall computational framework is complex but reason-able. What about the computational complexity and the image sample frequency?        Your series  of simulations and experiments are all done within simple indoor situations.  What about outdoor appliations?

Why not applying dynamic vision systems based on event cameras combined with RGB-D for ground truth?

Your paper is clearly written and considers at large the state of the art.  The novelty of the contribution is rated to be incremental.    There exists a large bibliography of related work on RGB 3D-SlAM. A comparison with related work would enhance the quality of the paper.

The paper should be shortened if possible. 
